# Disruption of Circadian Rhythms: A Crucial Factor in the Etiology of Infertility

**DOI:** 10.3390/ijms21113943

**Published:** 2020-05-30

**Authors:** Francesca Sciarra, Edoardo Franceschini, Federica Campolo, Daniele Gianfrilli, Francesco Pallotti, Donatella Paoli, Andrea M. Isidori, Mary Anna Venneri

**Affiliations:** 1Molecular and Clinical Endocrinology Laboratory, Department of Experimental Medicine, Sapienza University of Rome, 00161 Rome, Italy; francesca.sciarra@uniroma1.it (F.S.); edoardo.franceschini@uniroma1.it (E.F.); federica.campolo@uniroma1.it (F.C.); daniele.gianfrilli@uniroma1.it (D.G.); andrea.isidori@uniroma1.it (A.M.I.); 2Laboratory of Seminology–Sperm Bank “Loredana Gandini”, Department of Experimental Medicine, “Sapienza” University of Rome, 00161 Rome, Italy; francesco.pallotti@uniroma1.it (F.P.); donatella.paoli@uniroma1.it (D.P.)

**Keywords:** hormones regulation, clock genes, fertility, reproduction, spermatogenesis

## Abstract

Infertility represents a growing health problem in industrialized countries. Thus, a greater understanding of the molecular networks involved in this disease could be critical for the development of new therapies. A recent finding revealed that circadian rhythmicity disruption is one of the main causes of poor reproductive outcome. The circadian clock system beats circadian rhythms and modulates several physiological functions such as the sleep-wake cycle, body temperature, heart rate, and hormones secretion, all of which enable the body to function in response to a 24 h cycle. This intricated machinery is driven by specific genes, called “clock genes” that fine-tune body homeostasis. Stress of modern lifestyle can determine changes in hormone secretion, favoring the onset of infertility-related conditions that might reflect disfunctions within the hypothalamic–pituitary–gonadal axis. Consequently, the loss of rhythmicity in the suprachiasmatic nuclei might affect pulsatile sexual hormones release. Herein, we provide an overview of the recent findings, in both animal models and humans, about how fertility is influenced by circadian rhythm. In addition, we explore the complex interaction among hormones, fertility and the circadian clock. A deeper analysis of these interactions might lead to novel insights that could ameliorate the therapeutic management of infertility and related disorders.

## 1. Introduction

The term ‘’circadian rhythm”, derived from Latin “circa dies”, is used to describe the daily oscillations in gene expression, metabolism, activity patterns and serum hormone levels that occur across 24 h. These physiological processes represent an ubiquitous feature in living organisms, from cyanobacteria to humans, and are finely regulated by the suprachiasmatic nuclei (SCN) of the anterior hypothalamus [1]. In mammals, circadian timing system is generated at cellular levels through a series of interlocked positive and negative transcription/translation feedback loops, leading to the expression of circadian rhythms. The positive loop is composed by Brain and Muscle ARNT-like 1 (*BMAL1*) gene and Circadian Locomotor Output Cycles Kaput (*CLOCK*) gene, which encode for BMAL1 and CLOCK proteins, respectively. These proteins form an heterodimer, which finally binds itself to the a cyclic AMP response element (CRE)/E-box element (E-box) of Period (*PER*), Cryptochrome (*CRY*) genes, promoting their transcription [2]. Similar to their predecessors BMAL1 and CLOCK, also PER and CRY are able to dimerize, bind to Casein Kinase 1 (CK1), and finally move into the cell nucleus, where they binds to the BMAL1/CLOCK dimer, inhibiting its transcription “closing” the loop (negative loops) [3,4]. In a second transcriptional loop, CLOCK/BMAL1 activate the transcription of genes for the nuclear receptors *REV-ERB,* also known as *NR1D1* (nuclear receptor subfamily 1, group D, member 1). These proteins compete with the retinoic acid-related orphan receptors (ROR) for binding sites ROR-binding elements (RORE) on the *BMAL1* gene, providing both positive (ROR) and negative (REV-ERB) transcriptional regulation [4] (Figure 1).

Many data coming from animal studies have demonstrated the role of clock genes and clock-related genes in the regulation of both male and female fertility [5,6]. Indeed, alterations of biological rhythms and disrupted functions of the circadian clocks have been demonstrated to negatively impact reproductive capacity [7]. It is well known that physiological processes governing fertility need to be appropriately tight-timed orchestrated with the external environment to ensure reproductive success. Thus, a fine circadian regulation of reproductive hormones is mandatory for fertility both in males and females. It is interesting to note that the regulation of the estrus cycle, luteinizing hormone (LH) surge, sperm production and maturation, and the timing of insemination and fertilization are regulated by clock genes [6,8]. Furthermore, the timing of peripheral biological rhythm patterns is synchronized with circadian oscillation of melatonin and cortisol [9,10]. Changes in their circulating levels can indirectly impair reproduction, in which proper levels of glucocorticoids are required for normal gonadal function [11].

The aim of this review is to highlight most recent findings on the network connections among circadian rhythms, hormones and fertility.

## 2. How Fertility Is Influenced by Hormones and Clock Genes?

Originally, the molecular mechanism governing clock gene machinery was identified in the cells of a large number of tissues from several species. Afterwards, some studies demonstrated that clock machinery also had an influence on fertility and reproductive success [12,13]. This interaction is not only one-way, since fertility hormones can also influence clock-gene expression [6], thus indicating a complex network of interactions [14,15].

### 2.1. Gonadotropins

Fertility is finely regulated by the hypothalamic–pituitary–gonadal axis (HPG axis) and by two hypothalamic neuronal populations, the Kisspeptin neurons and the gonadotropin-releasing hormone (GnRH) neurons [16,17]. The Kisspeptin neurons, located in the anterior ventral periventricular area, are involved in the LH surge while the hypothalamic Kisspeptin neuron population, located in the arcuate nucleus, brings metabolic status information to the HPG axis and released GnRH [18,19]. The GnRH is released in a pulsatile manner by the hypothalamus and acts on the anterior pituitary lobe (Adeno-Hypophysis), regulating the production of gonadotropins and finally releasing follicle-stimulating hormone (FSH) and LH into the bloodstream. In males, LH is mandatory for testosterone production while FSH is involved in sperm production [20,21]. In females, FSH and LH are both involved in the production of steroid hormones (estrogens) by ovarian follicles [22]. Furthermore, the peak of the LH induces the continuous gene and protein expression of BMAL1 in the mouse ovary [23]. The impairment of FSH signaling results in poor spermatogenesis and subfertility in male [20] (Table 1). Moreover, since Kisspeptin signaling is mandatory for the production and activity of Leydig cells, germ cells progression and sperm functions, an alteration of this signaling causes hypogonadotropic hypogonadism [24]

### 2.2. Estrogens and Androgens

The existence of a link between circadian rhythms and estrogen has been clearly demonstrated [38,39] and a recent study showed that most of female fertility hormones display circadian rhythmicity [25]. During the follicular phase, FSH, LH, estrogen, progesterone, and sex hormone binding-globulin (SHBG) show a peculiar rhythmicity [25] (Table 1). In contrast, during the luteal phase, only FSH and SHBG display such rhythmicity. In females, estrogens stimulate LH surge [22] and determine the appearance of secondary sexual characteristics [28]. It is known that the expression of both *Per1* [40] and *Per2* [41] is modulated by estrogens. Importantly, a comprehensive study demonstrates that *Clock* gene is able to influence the activity of the estrogen receptor alpha (ERα), by regulating its transcriptional activity [42]. Furthermore, the link between the circadian system and estrogen synthesis is further strengthened by the discovery that ER receptors are expressed in the SCN [43]. Not only estrogens, but also androgens exert some effects on fertility. Androgens levels show a diurnal oscillation with morning peak levels [26]. In males, androgens are necessary for the development and maintenance of secondary sexual characteristics, libido, growth, prevention of osteoporosis, and spermatogenesis [29]. In females, LH regulates ovarian androgens production [26]. Female hyperandrogenemia (a common feature of the polycystic ovary syndrome, PCOS) is accompanied by an excessive production of androgens that affect *Clock* gene expression in ovarian rat follicles [44,45]. Metabolic syndrome and obesity in PCOS patients are commonly associated with a decline in reproductive function, disrupted reproductive cycles and attenuated gonadotropins secretion [46,47]. Furthermore, androgens display direct and tissue-specific effects on *Per2* gene expression that may account for the effects on the developmental program of the timing system [48].

### 2.3. Glucocorticoids

Glucocorticoids (GCs) are produced by adrenal glands and are synchronizers of endogenous clocks [9,10]. Their regulation displays a diurnal release pattern, with peak levels at the beginning of the activity phase [31] (Table 1). It was demonstrated that an increase of GCs levels occurring during a stress-mediated response [49] or due to GCs exogenous administration could severely impair different organs including immune [50,51] and reproductive systems [52]. These effects occur through an action on different hypothalamic–pituitary–adrenal (HPA) axis compartments: on the hypothalamus, inhibiting GnRH release; and on the pituitary, influencing gonadotropins synthesis and release [33]. GCs influence ovarian function also indirectly, by altering the levels of circulating gonadotropins and affecting levels of metabolic hormones and growth factors. Some recent discoveries point out that GCs can indirectly cause disruption of ovarian cyclicity, both through the inhibition of Kisspeptin neurons [53] and the stimulation of gonadotropin-inhibitory hormone (GnIH) [54]. On the pituitary gland, GCs inhibit the synthesis and release of LH and FSH, while on testis/ovary, they directly inhibit steroidogenesis and/or gametogenesis and induce apoptosis [55,56].

GCs exert their effect through the binding on glucocorticoid receptor (GR), which regulates the expression of glucocorticoids-responsive genes. The exogenous administration of GCs in female mice causes multiple damages on reproductive tissues [57]. In the uterus, it was described that an increase of GCs causes a reduction of embryos implantation rate [58]. Ovaries exposed to GCs treatment undergo several damages such as steroidogenesis alteration, histological impairment [59], and reduction of the total number of germ cells and ovarian volume [60]. In oocytes, GCs impair their reproductive competence, inducing oocytes apoptosis [61]. GCs are also able to compromise placenta integrity, decreasing its weight [62] and size together with a size reduction of newborn mice [63]. (Figure 2).

In male mice, adrenalectomy proved that the homeostatic level of hormones, including GCs, is essential to ensure a correct spermatogenesis, sperm maturation and steroidogenesis [32]. In male rats prenatally exposed to GCs, steroidogenesis is disrupted with a reduction in FSH and testosterone levels and a consequently altered sperm quality (decreased sperm concentration, motility and abnormal morphology). The exposure to betamethasone (BX) causes abnormalities in testes and epididymis, such as a reduction of anogenital distance and the alteration of other histo-morphometric parameters [64]. Moreover, it was demonstrated that not only the F1 generation, with fathers exposed prenatally to the BX, but also the F2 generation displays fertility damage [65]. In order to determine circadian rhythmicity in mouse testes, Chen and co-workers analyzed the expression levels of several circadian clock genes, including *Bmal1*, *Per1*, *Per2,* and *Cry1*. The expression of these genes was found to be arrhythmic in testes. They also demonstrated that *Bmal1* transcript levels and steroidogenic-related genes increased in Leydig cells after treatment with dexamethasone, indicating that circadian clockwork in Leydig cells may play a functional role in the control of testosterone secretion [66].

Regarding human male fertility, many evidences suggest that stress conditions (both physical and psychological) are followed by physiological responses that can impair reproductive functions [67]. In order to test the use of synthetic GCs in the management of male infertility, Prednisone has recently been tested in the treatment of accessory glands inflammation patients with oligospermia. GCs administration causes a considerable improvement in the sperm parameters of these patients, reducing inflammation [68]. However, other studies reported opposite results [69], rendering this question still open. Some trials have also shown that synthetic GCs are able to trigger a disruption in reproductive hormone levels [70], even if there is a lack of studies on this topic. Finally, a discrete number of observational studies suggested that testosterone levels of untreated controls were lowered by GCs [71]. However, randomized controlled trials are still missing.

### 2.4. Melatonin

Melatonin is a neurohormone secreted by the pineal gland whose secretion is regulated both by dark–light and seasonal cycles that exert pleiotropic actions like growth, development, senescence, and function of cells and organs in both male and female reproductive systems [34] (Table 1).

Human melatonin receptors have been localized in different cells like granulosa cells from preovulatory follicles [72] and in spermatozoa [73]. Melatonin levels have been identified to be lower in semen samples with low and poor fertilization rates [72] and in women with idiopathic infertility [74]. In stress-induced women, in which melatonin suppression is present, the circadian alteration affects fertility and fetus development, increasing the risk of miscarriage, premature birth and low birth weight [75]. It is also known that light emitted from electronic screen devices suppresses melatonin secretion that is linked with a decline of sperm motility and concentration in men [76]. Melatonin is important in the development of the fetal circadian clock, the neurological and endocrine systems, and to protect the embryo/fetus from metabolic stresses that can cause damage to the growing pregnancy. Mothers treated with melatonin can reset and reverse the fetal clock via the adrenal gland [37]. In addition, melatonin has been shown to preserve mouse spermatogenesis and reduce levels of free radicals, protecting sperm from oxidative damage [35,36]. Moreover, melatonin levels may influence the in vitro fertilization outcomes since its levels positively correlate with antral follicle count [77] and could therefore predict embryo quality [78]. However, some data suggest that an excess of melatonin leads to infertility [79]. Indeed, a reduction of melatonin levels can lead to precocious puberty, by affecting its onset [80]. Elevated melatonin levels have been associated with hypogonadism and amenorrhea in women [81,82] and with oligospermia or azoospermia in men [83].

## 3. Genetic Models of Clock Genes and Fertility

### 3.1. Female Fertility

All the information on the role of circadian rhythm in fertility come from knockout mouse studies (Table 2). *Per*1 and *Per*2 knockout mice are characterized by a marked decrease in reproductive rate as a consequence of irregularity and acyclicity of the estrous cycle [84,85] (Figure 3). The fertility of young *Per1* and *Per2* knockout mice does not differ from *wild-type* mice during mild-age, highlighting that *Per1*-*Per2* absence impairs infertility only in aged mice [84]. It was also demonstrated that *Per1*-*Per2* double knockout mice had a premature depletion of the ovarian follicular reserve, with a consequent decline in reproductive capacity [86]. To deeply understand the impact of clock genes on fertility, a transgenic mouse model of *Clock* gene was generated (*Clock^Δ19/^****^Δ19^******)***. These mice are characterized by a defective form of CLOCK protein, able to produce the BMAL1-CLOCK dimer, but unable to regulates *Per* and *Cry* transcription [87]. In addition to the loss of circadian rhythmicity, they also show higher rate of pregnancy failure [88].

*Bmal1* knockout mice, although able to ovulate, exhibited delayed puberty, irregular estrous cycles and smaller ovaries and uterus [90,92,93] (Figure 3). *Bmal1* absence negatively affects progesterone levels [91], causing embryo implantation failure [90]. This detrimental effect was explained with a disrupted expression of the *StAR* gene, critical for progesterone production regulation. In newborn *StAR* knockout mice, the adrenal glands lacked their normal cellular architecture and are characterized by high lipid depots. In contrast, testes contained only scattered lipid depots, while the ovaries appeared completely normal. According to this model, mice initially retain some capacity for StAR-independent steroidogenesis; thereafter, progressive lipid accumulation in steroidogenic cells, after hormone stimulation, kills the cells and completely abrogates steroidogenic capacity [91]. This model proves the interlocking and synergistic network of the circadian clock and reproductive systems. *Bmal1* knockout and *Clock^∆19/^****^Δ19^*** transgenic mice display undetectable LH levels [94] but still undergo ovulation and could become pregnant. This suggests that the circadian system was essential in gating the LH surge but was not required for successful ovulation [6]. Under certain circumstances, transgenic mice lacking LH pulsatility still had ovulation without the classical LH surge [95].

Chu and colleagues have recently shown that leptin receptor (LepR) is required for female fertility. Interestingly, *Bmal1* knockout mice showed a reduced expression of LepR and its ligand, reducing also estrogens concentration in granulosa cells. These results suggest that estrogens synthesis is regulated by *Bmal1* through the Leptin–LepR pathway [96].

Cholesterol represents an essential substrate for steroid hormone synthesis and genetic mutations altering synthesis and function of proteins involved in cholesterol uptake and mobilization from stored intracellular pools, significantly impact fertility [97].

To highlight the link among cholesterol, fertility and clock genes, an aromatase knockout mouse model ArKO was created [98]. This mutant mouse lacking functional aromatase cytochrome P450 gene (*Cyp19)* was unable to synthesize endogenous estrogens and, thus, ensure a correct ovulation. These mice had increased levels of testosterone, FSH and LH and are infertile due to the disruption of folliculogenesis and the absence of corpora lutea [99].

In order to better highlight the link among cholesterol, fertility and circadian regulation, a mutant mouse for *Clock* gene was generated [100]. These mice display reduced and arrhythmic expression levels of the enzyme 3-hydroxy-3-methylglutaryl-CoA reductase *(Hmgcr*), that regulates cholesterol synthesis in the liver, facilitating accumulation of cholesterol in this organ [100].

It is therefore clear that most of the factors involved in fertility are influenced, or influence themselves, the biological clock thus constituting a complex “network” of mutual interactions.

### 3.2. Male Fertility

Circadian genes expression in mouse testes, including the expression of *Per1, Per2*, *Bmal1, Clock*, and *Cry1*, is arrhythmic, suggesting that *clock* genes have non-circadian functions in spermatogenesis [101]. Instead, *clock* genes are rhythmically expressed in extra testicular ducts and accessory cells of mouse testis [102].

*Clock^Δ19/^****^Δ19^*** transgenic mice show slightly decreased male fertility and reduced newborn mice size [88], while canonical *Clock* knockout spermatozoa display lower in vitro fertilization, impact on blastula formation, and also lower acrosin (a sperm proteinase essential for fertilization) activity, through up-regulation of SERPINA3K [103,104] (Table 3).

As for female gametes, also in spermatozoa, *Bmal1* knockout determines a decrease of the expression of the *StAR* gene and related proteins [92]. *Bmal1* knockout mice are infertile and display low testosterone levels and high serum levels of LH, suggesting a Leydig cell dysfunction. It is well established that in these cells BMAL1 is expressed rhythmically, allowing to postulate a role in testosterone production [92] (Figure 3). A recent study also highlighted that the circadian rhythm of the Leydig cells endocrine functions decreases during aging [105].

The testicular function was studied in *Cry1* knockout mice [106]. *Cry1* was expressed in Sertoli cells, spermatogonia, spermatocytes and in the interstitium. These mice showed no phenotypic abnormalities, with similar serum and intratesticular testosterone levels to *wild-type* mice and normal Sertoli and Leydig cells [107]. However, there was an increased number of degenerated and apoptotic germ cells in the testis corresponding to lower epididymal sperm counts and testicular cell apoptosis. In addition, there was an upregulation of *Per2* levels in the testis respect to *wild-type* mice, whereas no differences were observed in the expression of other clock genes [106] (Figure 3).

### 3.3. Circadian Clock and Sexual Development

Sexual development processes are not exclusively under endocrine control but also own a circadian regulation level [38,44]. Reproductive organs’ circadian regulation influences many sexual developmental processes, like gonadotropin secretion timing, which has been demonstrated to answer to a circadian regulation itself, both regarding GnRH release [108] and gonadotropins secretion [109]. In addition, also the ovulation process was recently identified as having a circadian component, essential for its correct regulation [32].

The age of menarche represents the hallmark of puberty in females. It varies widely between individuals, is considered an heritable trait, and an alteration of its timing has been associated with risks for obesity, type 2 diabetes, cardiovascular disease, and breast cancer. Puberty in human and animals depends on a complex hormonal regulation [110,111]. Moreover, it was demonstrated that girls are more susceptible than boys to develop precocious puberty [112,113]. A strong association between puberty timing and risks to develop breast and endometrial cancers in women and prostate cancer in men has been reported [114]. The timing of menarche can also be influenced by light perception: In fact, in blind women, menarche was reported to be earlier with no differences in menopause when compared to sighted women [115].

Also the microbiome has a role in sex-specific diurnal rhythms. In fact, the absence of microbiome leads to an altered sexual development and growth hormone secretion. The composition and metabolic activity of commensal bacteria show diurnal variations that depend on the circadian clock [116] and vice versa the microbiota can interfere the clock gene expression in the host [117].

In germ-free mouse model, the resulting feminization of male and masculinization of female is likely caused by altered sexual development and growth hormone secretion, associated with differential activation of xenobiotic receptors [116].

Despite this interesting evidence of the circadian component in the modulation of sexual development, the role of clock genes in these processes is still understudied.

## 4. Effect of Clock Gene Mutation in Humans

Studies with mouse models clearly showed how clock genes can influence fertility, and vice versa. In humans there are mainly indirect evidences about this reciprocal influence in both sexes.

The women shift work influence on circadian regulation is considered one of the main factors causing prolonged waiting time to achieve pregnancy [118], suggesting that the sleep and inherent circadian rhythm disturbances of shift work could lead to menstrual irregularities due to altered levels of FSH, LH and prolactin [12,14,119]. Expression of *CLOCK* genes (*CLOCK, BMAL1, PER, CRY*) was analyzed in the full-term human placenta, indicating that placenta works as a peripheral circadian oscillator [37]. Moreover, some studies analyzed if progesterone could have any influence on clock machinery functioning. In details, *PER1* levels increase and remain high during the decidualization of the human endometrium. The progesterone receptor activated *PER1* transcription by directly binding to its promoter ensuring a correct decidualization [120] (Table 4).

There are many evidences of clock genes rhythm in human ovary. *CLOCK* was detected in the granulosa cells of the dominant antral follicle of in vitro fertilization patient, however, its expression was undetectable in preantral and primary ovarian follicles [123]. Even if *CLOCK* and *PER1* were expressed, their mRNA expression levels undergo a decrease with the increase of patient age [124]. Furthermore, it was shown that the expression of *PER2* and *STAR* can be induced in vitro by testosterone, outlining a potential link between *CLOCK* genes and steroidogenesis, a relation that was highlighted in experimental mouse models [98].

Another recent study investigated the possible link between the expression of circadian rhythm in decidual endometrial cell tissues and the recurrent miscarriage (RM), defined as two or more consecutive losses of a clinical intrauterine pregnancy. With the use of human endometrial stromal cells (HESCs), isolated from decidua of first-trimester pregnancies, it was demonstrated that the expression of *BMAL1* was reduced in the endometrial tissues of RM patients [121]. The same study verified that *BMAL1* absence in the endometrial cells results in damaged decidualization and in an aberrant trophoblastic invasion, predisposing to RM. Furthermore, a single-nucleotide *BMAL1* polymorphism (rs2278TT749 TT) has been found, and it has been associated both with a great number of miscarriages but also with an increased number of pregnancies [122]. *CLOCK* gene was detected in human healthy fetuses and reached a peak of expression during the 6th week of gestation. Its expression was instead abnormal in spontaneous miscarriage fetuses, compared to healthy fetuses [125].

The same indication comes also from a study regarding the human Han-Chinese population, which has been demonstrated to display some *CLOCK* gene single nucleotide-polymorphisms that is associated with idiopathic infertility [126]. Going deeply, the rs1801260 polymorphism associates with normal seminal parameters, while rs3817444 associates with both normal and abnormal seminal parameters [126]. Another study within this population highlighted the connection between *CLOCK* gene polymorphism and semen quality in idiopathic infertility [107]. Heterozygotes and homozygotes males (TC and CC) for a C allele variant at rs3749474 show a significant reduction in seminal volume, compared to TT genotype, and in addition, CC homozygotes have also both significantly lower concentration and sperm motility. Another polymorphism, the rs1801260, in heterozygotes TC genotype, shows a significantly lower motility compared to the TT genotype [127]. In a recent study, a homozygous mutation was identified as likely disease causing in the *NPAS2* gene in a family of three brothers from Turkey, affected by Non-Obstructive Azoospermia [128] (Table 5).

All these works highlight a potential connection between the *Clock* gene and human male fertility, but more experimental works and efforts are necessary to discover more about this topic.

## 5. Conclusions

Infertility represents one of the most important health problems following cancer and cardiovascular diseases. Few studies have explored the relation between circadian rhythms and reproduction and most of them have been made in rodents. However, the discovery of functional clock machinery in many reproductive tissues and the development of clock gene knockout mouse models have revealed the pivotal role of such genes in orchestrating reproductive processes in mammals. Clock genes affect infertility, producing low levels of sex hormones, causing embryo implantation failure and reducing newborn size both in mouse models and in shift-working women.

In this review, we assessed the connection between hormones and fertility from the viewpoint of the circadian system. Although we may not cover the whole field of disease phenotypes, we believe this review highlights the intimacy with which these three physiological aspects interact to maintain homeostasis.

Understanding how the circadian rhythms work, and how genes make them run, is becoming more and more important for human infertility. A tight net of interactors, among which clock genes and hormones are the major players, define a complex network. However, more studies are required to better fix the topic, especially regarding humans. In the years to come, the knowledge of the infertility problem could be increasingly linked to the understanding of this vast network of interactions. Chrono-therapeutical strategies that reset or modify the biological clock may contribute to restore the internal synchrony and thus counteract pathological symptoms of infertility.

## Figures and Tables

**Figure 1 ijms-21-03943-f001:**
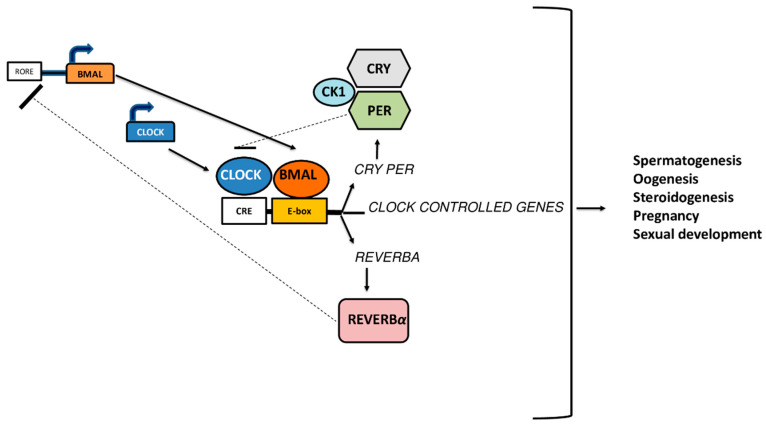
Autoregulatory feedback loop of clock-specific gene expressions that are involved in fertility processes.

**Figure 2 ijms-21-03943-f002:**
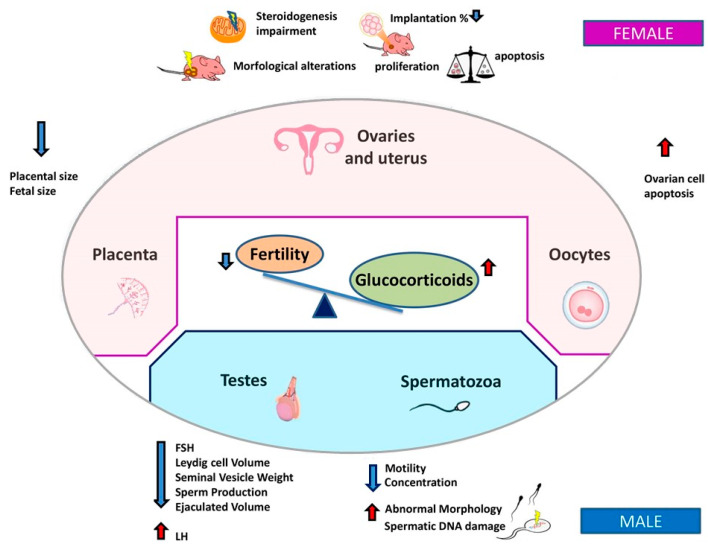
Infertility is associated with unbalanced GCs levels. The reciprocal relationship between the disruption of circadian rhythms of GCs and fertility may be either the cause or the effect of female and male infertility.

**Figure 3 ijms-21-03943-f003:**
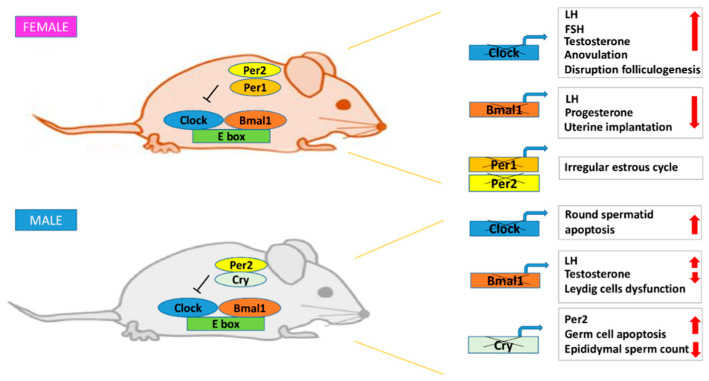
Schematic representation of infertility complications due to clock gene disruption on female and male knockout mice.

**Table 1 ijms-21-03943-t001:** Rhythmicity and physiological effects of hormones in male and female fertility.

Hormones	Rhythmicity		Effects on Male	Effects on Female
FSH	24 h circadian rhythm during follicular phase [25]	24 h circadian rhythm during luteal phase [25]	Sertoli cell tropism and sperm production [20]	Stimulation of estrogens production by ovarian granulosa cells [22]
LH	24 h circadian rhythm during follicular phase [25]	No circadian rhythm in luteal phase [25]	Stimulation of testosterone production by Leydig cells [21]	Stimulation of estrogens production by ovarian granulosa cells [22]
				Regulation of theca cells androgen production [26]
Estrogens	24 h circadian rhythm during follicular phase [25]	No circadian rhythm in luteal phase [25]	Regulation of ductal and epididymal function [27]	Development and maintenance of secondary sexual characteristics [28]
Androgens	24 h circadian rhythm with a peak in the early morning [26]		Development and maintenance of secondary sexual characteristics [29]	Control of growing follicles [30]
Glucocorticoids	24 h circadian rhythm with a peak in the morning [31]		Promotion of sperm maturation and steroidogenesis [32]	Regulation of fetal growth and development [33]
Melatonin	24 h circadian rhythm with a peak in the night [34]		Preservation of spermatogenesis [35,36]	Control of neurological and endocrine systems development [37]
			Reduction of free radicals protecting sperm from oxidative damage [35,36]	Protection of the embryo/fetus from metabolic stress [37]

**Table 2 ijms-21-03943-t002:** Effect of disrupted genes in female mice mutant models.

Disrupted Genes	Effects	Ref
*Per1, Per2*	Significant decrease of ovarian follicles in aged mice	[86]
	Accelerated reproductive aging	[86]
*Clock^Δ19/Δ19^*	Higher rate of pregnancy failure in aged mice	[89]
*Bmal1*	Delayed puberty	[90]
	Irregular estrous cycles	[90]
	Smaller ovaries and uterus	[90]
	Disrupted StAR gene expression	[91]
	Lower progesterone levels	[91]

**Table 3 ijms-21-03943-t003:** Effect of disrupted genes in male mutant mice models.

Disrupted Genes	Effects	Ref
*Clock*	Significant fertility reduction	[103]
	Lower in vitro fertility rate	[103]
	Lower blastula formation rate	[103]
	Lower acrosin activity	[103]
*Clock^Δ19/Δ19^*	Mild sperm fertility in aged mice	[88]
*Cry*	Increase apoptosis of germ cells	[106]
	Lower epididimal sperm count	[106]
*Bmal1*	Total infertility	[92]
	Disrupted StAR gene expression	[92]
	Leydig cell impairment	[105]

**Table 4 ijms-21-03943-t004:** Effect of mutated genes on human female fertility.

Mutated Genes	Effects	Ref
*PER1*	Attenuated human endometrial decidual transformation	[120]
*BMAL1*	Damaged decidualization	[121]
	Aberrant trophoblastic invasion	[121]
*BMAL1* polymorphism *rs2278TT749*	Associated both with a great number of miscarriages but also with an increased number of pregnancies	[122]

**Table 5 ijms-21-03943-t005:** Effect of mutated genes on human male fertility.

Mutated Genes	Effects	Ref
CLOCK polymorphism rs1801260	Normal seminal parameters	[126]
CLOCK polymorphism rs3817444	Normal and abnormal seminal parameters	[126]
CLOCK polymorphism rs1801260 TC genotype	Lower motility compared to the TT genotype	[127]
CLOCK polymorphism rs3749474 CC genotype	Seminal volume reduction, lower concentration and sperm motility compared to TT genotype	[127]

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
