# Peer review of "Disruption of Circadian Rhythms: A Crucial Factor in the Etiology of Infertility"

_ijms, 2020, doi:10.3390/ijms21113943_

Round 1

Reviewer 1 Report

This review article by Sciarra and colleagues represents a comprehensive and interesting review about the connection between the circadian clock and the reproductive system. Nevertheless, a few points are problematic and need clarification:

- On line 192, in the chapter related to melatonin, authors only discussed the positive effect of melatonin on reproduction. However, excess melatonin leads to infertility (1)! This chapter is therefore misleading and needs rewriting with the addition of the negative effect of melatonin on reproduction.

- On line 239, authors referred to a “Stard4” ko without any additional explanation. It would be great to discuss what interesting discovery is associated with this model.

- On line 332, authors talked about a “functional molecular clock machinery in reproductive tissues”. However, as properly stated in the text and reported by others (2), there is no circadian rhythmicity in testis!

- One important element that is missing in this review is the link between the circadian clock and sexual development. Mouse models deficient for clock genes present a defective sexual maturation (3). Moreover, light perception (4) and polymorphisms in BMAL1 are associated with age of menarche in human (5-7). It could be therefore interesting to add a paragraph related to circadian clock and sexual development.

- Annotations of genes in mouse and human need to be corrected.

References

  1. M. M. Macchi, J. N. Bruce, Human pineal physiology and functional significance of melatonin. Front Neuroendocrinol 25, 177-195 (2004).
  2. D. Morse, N. Cermakian, S. Brancorsini, M. Parvinen, P. Sassone-Corsi, No Circadian Rhythms in Testis: Period1 Expression Is Clock Independent and Developmentally Regulated in the Mouse. Mol. Endocrinol. 17, 141-151 (2003).
  3. B. D. Weger et al., The Mouse Microbiome Is Required for Sex-Specific Diurnal Rhythms of Gene Expression and Metabolism. Cell Metab 29, 362-382.e368 (2019).
  4. E. E. Flynn-Evans, R. G. Stevens, H. Tabandeh, E. S. Schernhammer, S. W. Lockley, Effect of Light Perception on Menarche in Blind Women. Ophthalmic Epidemiol 16, 243-248 (2009).
  5. F. R. Day et al., Genomic analyses identify hundreds of variants associated with age at menarche and support a role for puberty timing in cancer risk. Nat Genet 49, 834-841 (2017).
  6. C. E. Elks et al., Thirty new loci for age at menarche identified by a meta-analysis of genome-wide association studies. Nat Genet 42, 1077-1085 (2010).
  7. J. R. B. Perry et al., Parent-of-origin-specific allelic associations among 106 genomic loci for age at menarche. Nature 514, 92-97 (2014).

Author Response

We would like to thank the editor and reviewers for their time and effort spent on reviewing our manuscript. They have given us many helpful suggestions and remarks which we believe have improved the quality of the manuscript. All of the comments and suggestions have been addressed and incorporated in the revised version.

Reviewer 1 general comments:

This review article by Sciarra and colleagues represents a comprehensive and interesting review about the connection between the circadian clock and the reproductive system. Nevertheless, a few points are problematic and need clarification:

Reviewer 1

specific comments:

author response and changes made

page number in revised paper where the change can be found

1

On line 192, in the chapter related to melatonin, authors only discussed the positive effect of melatonin on reproduction. However, excess melatonin leads to infertility (1)! This chapter is therefore misleading and needs rewriting with the addition of the negative effect of melatonin on reproduction.

We thank the reviewer and we agree that the chapter related to melatonin was not exhaustively described. We amended the manuscript and we also highlighted that the excess melatonin leads to infertility.

On lines 183-187

2

On line 239, authors referred to a “Stard4” ko without any additional explanation. It would be great to discuss what interesting discovery is associated with this model.

We thank the reviewer for the useful comment and we have included a brief description of ''StAR'' KO mice.

On lines 212-217

3

On line 332, authors talked about a “functional molecular clock machinery in reproductive tissues”. However, as properly stated in the text and reported by others (2), there is no circadian rhythmicity in testis!

We thank the reviewer and we have corrected mistakes in the text.

On lines 245-248

4

One important element that is missing in this review is the link between the circadian clock and sexual development. Mouse models deficient for clock genes present a defective sexual maturation (3). Moreover, light perception (4) and polymorphisms in BMAL1 are associated with age of menarche in human (5-7). It could be therefore interesting to add a paragraph related to circadian clock and sexual development.

We thank the reviewer for comments and we  have included this section as requested.

On lines 270-295

5

Annotations of genes in mouse and human need to be corrected.

We thank the reviewer for the observations.

All along the manuscript

Reviewer 2 Report

Sciarra and colleagues have reviewed the literature on the connection between circadian rhythms, hormones and (in)fertility. The topic is interesting and worth reviewing, since there are not so many recent reviews about it.

Overall, I found the manuscript a bit confusing and not easy to read. Connections and explanations are missing in several occasions and the transition from one idea to the other is not always logic. There are also grammatic and formatting errors that need to be corrected. Figures and tables could also be improved.

Specific points:

- Introduction: a figure illustrating the feedback loops regulating specific gene expression and the general physiological consequences of activating/repressing these genes would help.

- Hormones part: this is the most confusing part, with too much and not well-connected information. For instance, male and female data are mixed in an unclear way. There is one figure, but only for glucocorticoids (is there any reason for this?). Maybe a table summing up the most important data for each hormone, with separated parts for males and females, would help. Data from KO animals should be moved to the genetic models part. All in all, this part needs to be re-written in a way that someone that is not familiar with the topic can understand it.

- Figure 2: Letters are too small

- Tables 1, 2, 3: Details are missing in the tables and the text. For instance: not everyone knows what IVF is, or what acrosin and its importance in sperm is. Table I, reference 106: how was “reproductive rate” measured? Table II, reference 119: how was the “decrease in fertility” assessed? Does (in)fertility in these tables referring to both males and females? Table 3: some of the readers may not know what decidua and decidualization are.

Author Response

Reply to referee

We would like to thank the editor and reviewers for their time and effort spent on reviewing our manuscript. They have given us many helpful suggestions and remarks which we believe have improved the quality of the manuscript. All of the comments and suggestions have been addressed and incorporated in the revised version.

Reviewer 2 general comments:

Sciarra and colleagues have reviewed the literature on the connection between circadian rhythms, hormones and (in)fertility. The topic is interesting and worth reviewing, since there are not so many recent reviews about it.

Overall, I found the manuscript a bit confusing and not easy to read. Connections and explanations are missing in several occasions and the transition from one idea to the other is not always logic. There are also grammatic and formatting errors that need to be corrected. Figures and tables could also be improved.

Reviewer 2

specific comments:

author response and changes made

page number in revised paper where the change can be found

1

Introduction: a figure illustrating the feedback loops regulating specific gene expression and the general physiological consequences of activating/repressing these genes would help.

We thank the reviewer. We provided a new Figure 1 to highlight the autoregulatory feedback loop of clock specific gene that are involved in fertility processes. In the previous version of the manuscript we had already shown the consequences of disruption of clock genes on female and male fertility in  Figure 2, now recall Figure 3.

On lines 61

2

Hormones part: this is the most confusing part, with too much and not well-connected information. For instance, male and female data are mixed in an unclear way. There is one figure, but only for glucocorticoids (is there any reason for this?). Maybe a table summing up the most important data for each hormone, with separated parts for males and females, would help. Data from KO animals should be moved to the genetic models part. All in all, this part needs to be re-written in a way that someone that is not familiar with the topic can understand it.

We thank the reviewer. We improved the hormones section as required. We separated parts for males and females, and we moved all information about KO mice in the genetic models paragraph. We added a new Table 1  to elucidate the rhythmicity and physiological effects of hormones in male and female fertility. About the decision to draw a figure about association of infertility with unbalanced GCs , we have tried to explain how relevant this hormone is in these disorders.

On lines 89; on line 189-295

3

Figure 2: Letters are too small

We thank the reviewer for the observations on figure 2. We increased the font size

On lines 204

4

Tables 1, 2, 3: Details are missing in the tables and the text. For instance: not everyone knows what IVF is, or what acrosin and its importance in sperm is. Table I, reference 106: how was “reproductive rate” measured? Table II, reference 119: how was the “decrease in fertility” assessed? Does (in)fertility in these tables referring to both males and females? Table 3: some of the readers may not know what decidua and decidualization are.

We thank the reviewer for useful comments and we substantiated all the observations.

On lines 202; On lines 260; On lines 309.